# Extended Cleavage Specificities of Rabbit and Guinea Pig Mast Cell Chymases: Two Highly Specific Leu-Ases

**DOI:** 10.3390/ijms20246340

**Published:** 2019-12-16

**Authors:** Yuan Zhongwei, Srinivas Akula, Zhirong Fu, Lawrence de Garavilla, Jukka Kervinen, Michael Thorpe, Lars Hellman

**Affiliations:** 1Department of Cell and Molecular Biology, Uppsala University, Uppsala, The Biomedical Center, Box 596, SE-751 24 Uppsala, Sweden; yuanzhongwei_08@163.com (Y.Z.); Srinivas.Akula@icm.uu.se (S.A.); fuzhirong.zju@gmail.com (Z.F.); getmeinahalfpipe@gmail.com (M.T.); 2GDL Pharmaceutical Consulting and Contracting, Downingtown, PA 19335, USA; ldegarav@aol.com; 3Tosoh Bioscience LLC, 3604 Horizon Drive, King of Prussia, PA 19406, USA; Jukka.Kervinen@tosoh.com

**Keywords:** mast cells, chymase, tryptase, cleavage specificity, Leu-ase, phage display

## Abstract

Serine proteases constitute the major protein content of mast cell (MC) secretory granules. These proteases can generally be subdivided into chymases and tryptases based on their primary cleavage specificity. Here, we presented the extended cleavage specificities of a rabbit β-chymase and a guinea pig α-chymase. Analyses by phage display screening and a panel of recombinant substrates showed a marked similarity in catalytic activity between the enzymes, both being strict Leu-ases (cleaving on the carboxyl side of Leu). Amino acid sequence alignment of a panel of mammalian chymotryptic MC proteases and 3D structural modeling identified an unusual residue in the rabbit enzyme at position 216 (Thr instead of more common Gly), which is most likely critical for the Leu-ase specificity. Almost all mammals studied, except rabbit and guinea pig, express classical chymotryptic enzymes with similarly extended specificities, indicating an important role of chymase in MC biology. The rabbit and guinea pig are the only two mammalian species currently known to lack a classical MC chymase. Key questions are now how this major difference affects their MC function, and if genes of other loci can rescue the loss of a chymotryptic activity in MCs of these two species.

## 1. Introduction

Mast cells (MCs) are innate hematopoietic cells distributed primarily at the interphase between the host and the environment. They are found in connective tissue and at mucosal surfaces, such as the intestinal or lung mucosa, and are often close to vascular and lymphatic vessels [1]. MCs can be triggered by several receptor-ligand pairs, including receptor-bound IgE and allergens, anaphylatoxins (C3a, C4a, and C5a), or substance P and their receptors. When triggered, the MCs release a number of physiologically potent mediators, including histamine, heparin, various proteases, prostaglandins, and leukotrienes [2]. Many of these inflammatory mediators are stored within cytoplasmic granules of the MC. Proteases constitute the major protein content of these granules. The majority of these proteases belong to the serine proteases of the trypsin/chymotrypsin family, where MCs store both tryptic and chymotryptic enzymes named tryptases (EC 3.4.21.59) and chymases (EC 3.4.21.39), respectively [3]. Most chymases cleave, preferably, after aromatic amino acids, including Phe, Tyr, and Trp, whereas tryptases prefer positively charged amino acids (Lys, Arg) at the P1 position of the substrate [4]. Phylogenetic analyses identify two distinctive subfamilies of chymases named α- and β-chymases. The genes encoding α-chymase are generally present as a single gene in nearly all species analyzed except for cows and sheep, which both have two very closely-related α-chymases, whereas β-chymases are mainly found in rodents, as well as in cats and dogs [5,6,7].

β-chymases are particularly interesting from both an immunological and evolutionary perspective. Analyses of the genome of a number of mammals show that β-chymases exist in the genome along with the α-chymase in all of the rodents analyzed, including the mouse, rat, golden hamster, Chinese hamster, and rabbit. Interestingly, the α-chymases in all rodents analyzed, except the guinea pig, have changed primary specificities from chymases to elastolytic activities (i.e., elastases), which are cleaving after aliphatic amino acids, such Val, Ile, and Ala, instead of after classic chymase aromatic amino acids [3].

Analysis of the β-chymases, rat mast cell proteases (rMCP-1 and rMCP-4), and mouse mast cell protease 4 (mMCP-4), by phage display shows that these proteases prefer Phe or Tyr at the P1 position, indicating that β-chymases became the primary chymotryptic enzymes in rodents when the function of the α-chymases changed to elastases [8,9]. Interestingly, β-chymase genes have also been identified in two other placental mammals: the cat and dog [5]. However, due to the uncertainty about the quality of the dog genome sequence, there is doubt concerning the existence of the dog β-chymase gene, as it was found in one genome sequence release and subsequently gone in the next. In the dog genome release from 2015, two chymases were present in the sequence: one α-chymase and one additional chymase more closely related to the β-chymases [5]. However, in the most recent release, this latter gene is missing. Attempts to clone the β-chymase cDNA by PCR amplification with specific primers designed from the 2015 genome sequence release, using RNA from different dog tissues have also failed, indicating that the gene may not be active and, therefore, should be considered a pseudogene [10]. However, a closely related β-chymase gene has also been identified in the cat genome with high similarity in the sequence, as well as at the same position in the chymase locus as the dog and rodent genes. We have also been able to clone a piece of the ‘β-chymase gene’ by PCR from wolf DNA, and this fragment was identical in sequence to the dog gene of the 2015 release [10]. For this reason, we are confident that the gene is present but simply lost in the genome assembly for unknown reasons. So far, no studies of the functions nor cleavage specificities of these genes have been performed. We hypothesized that the β-chymases are relatively old enzymes, dating back to the appearance of placental mammals. However, the β-chymase genes have been silenced or lost in some lineages, such as the primates [5].

The primary and extended cleavage specificities can provide clues to the potential targets of these enzymes. By identifying in vivo substrates of these enzymes, we also obtained information concerning their primary functions. Angiotensin I (Ang I) is one of the most likely in vivo substrates for the MC chymases. The human α-chymase (Cma1) can efficiently generate Ang II from Ang I by cleaving the peptide bond between Phe8 and His9 in Ang I [11]. Similarly, the dog α-chymase can also generate Ang II at a comparable efficiency to the human chymase [12]. The rat mast cell protease-1 (rMCP1), mouse mast cell protease-4 (mMCP4), and hamster chymase-1 (HAM1, MCP-1), which are all β-chymases, also generate Ang II. However, rMCP1 also has a tendency to degrade Ang II by cleavage at Tyr 4 in Ang II [3,11,13]. The fact that both α-chymases and β-chymases can generate Ang II in a distinct pathway from the traditional angiotensin-converting enzyme (ACE) indicates that the MC chymases are important Ang II converters [12].

Cytokines and chemokines are also potential targets. Analyses of the cleavage of 51 active recombinant cytokines and chemokines by the human MC chymase (HC) have shown that only three out of 51 of these proteins (IL-33, IL-18, IL-15) were substantially cleaved [14]. The cleavage activity of the IL-1–related alarmins IL-18 and IL-33, by both the HC and mMCP-4, also demonstrates the possibility for chymase to cleave cytokines or chemokines, and the cleavage activities might be similar among different species. Apart from cytokines and chemokines, both human and dog chymases can degrade fibronectin, indicating a role in tissue remodeling [15,16]. The number of potential substrates identified indicates that these enzymes may have a complex role in vivo, with several important functions in both immunity and tissue homeostasis.

By using a combination of a panel of different methods, including phylogenetic analyses, phage display, cleavage of recombinant and chromogenic substrates, alignments of primary structures, and 3D structural predictions, here we presented a detailed picture of the rabbit: Cma1-like protein concerning its enzymatic activity, substrate selectivity structure, and evolutionary relationship to other mammalian MC chymases. We also compared the rabbit enzyme with the guinea pig β-chymase, which showed a remarkably similar cleavage specificity as the rabbit enzyme, indicating convergent evolution by α- and β-chymases, highlighting a potentially new set of in vivo substrates.

## 2. Results

### 2.1. Phylogenetic and Primary Structure Analysis

A phylogenetic analysis of a panel of mammalian chymases was performed in order to compare these to the primary sequence of the rabbit Cma1-like and the guinea pig Cma1. The phylogenetic analysis showed that the rabbit Cma1-like clustered together with the β-chymases, which actually formed a separate branch representing the most divergent member of the different β-chymases (Figure 1A). Interestingly, the guinea pig enzyme was also the most divergent member of the different rodent α-chymases. It was found to be even more distant than the rabbit α-chymase in this analysis. Sequence alignment of a panel of mammalian α- and β-chymases showed that the rabbit Cma1 (an α-chymase) was most likely an inactive pseudogene due to the serine of the active site being mutated into a leucine (Figure 1B). Analysis of the rabbit β-chymase (the Cma1-like gene) also revealed one important difference compared to the other β-chymases: The predicted substrate pocket was lined by the three amino acids Ser189, Thr216, and Val226, whereas most other β-chymases had Ser189, Gly216, and Ala226 in these three positions. This difference in the amino acids might indicate that the rabbit Cma1-like might differ in the primary specificity from the majority of the other mammalian β-chymases (Figure 1B) [5]. Furthermore, the guinea pig Cma1 showed similar but not identical residues as the rabbit in these three positions, namely Ser189, Ala216, and Val226, indicating a similar primary specificity. The guinea pig enzyme has previously been shown to have a preference for leucine over aromatic amino acids in the P1 position of substrates, indicating that it is a relatively strict Leu-ase [17]. No analysis of the rabbit enzyme has previously been performed.

### 2.2. Purification and Activation of the Recombinant Rabbit and Guinea Pig Enzymes

To be able to perform functional studies of the rabbit and guinea pig enzymes, pure and active enzymes were required. The coding region for the guinea pig chymase was, therefore, inserted into the baculovirus vector pAcGP67B, expressed in baculovirus-infected insect cells, and purified, as previously described in detail [18]. Following purification, the fully mature enzyme (25–28 kDa) was ≥95% pure as determined by SDS-PAGE (Figure 2), and the correct N-terminus was confirmed by N-terminal sequencing. Rabbit Cma1-like was produced as a recombinant enzyme in a mammalian cell line, HEK293-EBNA. An N-terminal His-6 tag and an enterokinase site were added to enable purification using Ni-NTA agarose from the conditioned media, and the enterokinase site to remove the His-tag and obtain an enzymatically active protease. Following purification, the enzyme was activated by adding enterokinase for 5 h at 37 °C, which subsequently removed the His-6 tag and the enterokinase site. The cleavage resulted in a reduction in the size of the protein by ~2 kDa, which was confirmed by SDS-PAGE analysis (Figure 2).

### 2.3. Determination of the Extended Cleavage Specificity of the Rabbit Cma1-Like Enzyme by Substrate Phage Display

A T7 phage library was used to determine the extended cleavage specificity of the rabbit enzyme. The library used contained approximately 50 million phage clones. Each clone expresses a unique random sequence of nine amino acids followed by a His-6 tag to immobilize the phages onto a matrix, Ni^2^+ chelating agarose beads. Selection of the cleavage by the rabbit Cma1-like was performed for six rounds, resulting in approximately 100-fold more phages compared to negative control. One hundred and eleven individual phage colonies were picked, and a region of approximately 300 bp, including the coding region for the nine amino acid random region, was amplified by PCR. Ninety-six of the PCR products with the most distinct PCR bands were sent for sequencing. The sequencing results were translated into an amino acid sequence, and the nine random amino acids of each sequence were aligned by hand based on common sequence characteristics (Figure 3). The frequency of each amino acid in different positions (P5-P4′) was plotted as an Ice-Logo for visual representation (Figure 3) [21].

Rabbit Cma1-like showed a strict preference for Leu in the P1 position (Figure 3). A relatively equal distribution for three amino acids (Leu, Phe, and Asp) was observed for the P2 position, with only a few other amino acids in this position (Figure 3). Although several different amino acids were observed in the P1′ position, some preference for Ser, Met, Ala, and Val was observed. Additionally, a strong preference for aliphatic amino acids, both upstream and downstream of the cleavage site, was observed (Figure 3).

### 2.4. Verifying the Consensus Sequence by the Use of Recombinant Protein Substrates

To verify the sequences obtained from the phage display analysis, we used a new type of recombinant (protein) substrate. The consensus sequences obtained from the phage display analysis and several variants of this sequence were designed and produced as recombinant substrates in a two-thioredoxin (Trx) system. This system makes it possible to analyze preference for residues both N and C-terminal of the cleavage site in a very visual manner. Chromogenic substrates also often only contain the amino acids N terminally of the cleavage site and can, therefore, not be used to study the full extended specificity of a protease. A number of proteases that show high selectivity for amino acids both N-and C-terminal of the cleavage site can, therefore, not be analyzed with chromogenic substrates as then no cleavage can be detected with the chromogenic substrate, a phenomenon we have observed with several highly specific proteases. Such proteases can easily be studied with the 2-Trx substrates. The beauty with the system is also that it can be used with relatively inexpensive equipment and give results similar or even better than all other existing techniques to obtain quantitative information concerning the preference for amino acids at and around the actual cleavage site of an endopeptidase. Here, the aims were to confirm the results of the phage display and to determine the impact of individual amino acids in the different positions surrounding the cleavage site. We have produced more than 270 such 2x Trx substrates, and despite this, none of the existing substrates that were tested showed any detectable cleavage with the rabbit enzyme, as this enzyme had a quite different preference compared to other mammalian chymases. All substrates for the validation of the rabbit enzyme were, therefore, newly designed and produced. Double-stranded oligonucleotides encoding the consensus substrates and a number of variants of these substrates were designed, ordered, and ligated into the 2x Trx substrate vector. A His-6 tag was added in the C-terminal of the second Trx molecule to facilitate purification (Figure 4A). The vectors carrying the target sequences were transferred into *Escherichia coli* Rosetta gami for expression and purification. The purified 2x Trx proteins were then used to analyze the specificity of the rabbit Cma1-like protease (Figure 4B).

The analysis of the rabbit enzyme with the recombinant substrates confirmed the Leu-ase specificity observed from the phage display analysis. No cleavage was detected for substrate sequences lacking a Leu, which were intentionally designed as negative controls. This result demonstrated that rabbit Cma1-like had a strict preference for Leu in the P1 position. In addition, substrates with a Leu or Phe in the P2 position were cleaved with nearly equal efficiency (Figure 4C). The substrate with Asp in the P2 position was cleaved approximately three times less efficiently compared to the most efficient Leu-Leu or Phe-Leu substrates (Figure 4C). An Asp in the P3 position was also found to have a negative impact on the activity of the enzyme. Rabbit Cma1-like also preferred small hydrophobic amino acids, such as Val, in the P3 position (Figure 4C). Positively charged residues were very unfavorable when positioned just upstream of the cleavable Leu residue. Inserting Lys or Arg in the P2 position almost completely inhibited cleavage (Figure 4D). His and Pro in position P1′, immediately C-terminal of the cleavage site, also strongly inhibited cleavage (Figure 4E). The same effect was also observed for a Leu in the P4 and a His in the P3 position (Figure 4F).

The guinea pig chymase is the only protease that has previously been reported to have a similar cleavage specificity as the rabbit enzyme [17]. Due to the potential similarity in substrate specificity between the rabbit Cma1-like and the guinea pig Cma1, the guinea pig chymase was also analyzed with the same substrates as for rabbit Cma1-like (Figure 5). We had previously tried to determine the specificity of the guinea-pig enzyme with chromogenic substrates without success. Interestingly, using the 2x Trx substrates, the guinea pig Cma1 displayed almost identical cleavage specificity as the rabbit Cma1-like. The guinea pig chymase also preferred Leu in the P1 position, and substrates with Leu or Phe in the P2 position were cleaved with nearly equal efficiency (Figure 5A). An Asp in the P3 position showed lower activity and, similar to the rabbit enzyme, the guinea pig enzyme showed no cleavage of the substrate lacking a Leu (Figure 5A).

However, there were some important differences. The guinea pig enzyme was less affected by positively charged residues in the P2 position and by His in the P1′ position (Figure 5B,C). In contrast, a Pro in the P1′ position was similar to the rabbit enzyme, which almost completely inhibited cleavage (Figure 5C). However, a Leu in the P4 position was less detrimental to the cleavage compared to the rabbit enzyme (Figure 5D). Thus, despite the extended cleavage specificities being very similar between the two enzymes, there were subtle differences in the selectivity in certain positions of the substrates.

### 2.5. Structural Modeling of the Active Site of Rabbit Cma1-Like

To analyze the structural changes in the rabbit enzyme that may explain the transformation of the classical chymase into a highly specific Leu-ase, we performed structural modeling based on the primary structure of the rabbit β-chymase. X-ray crystallography analysis of the hamster chymase gene, HAM2 (hamster Cma1 in Figure 1), defines the positions 189, 190, 216, and 226 based on chymotrypsinogen numbering, to line the substrate pocket of the active site in all the active serine proteases [19]. For the enzymes with chymase activity, Ser_189_-Ala_190_-Gly_216_-Ala_226_, Thr_189_-Ala_190_-Gly_216_-Ala_226_, or Ala_189_-Ala_190_-Gly_216_-Ala_226_ are predicted to form the substrate pocket, whereas Asn_189_-Val_190_-Val_216_-Ala_226_ or Asn_189_-Val_190_-Val_216_-Ser_226_ form the pocket for elastases [22]. Interestingly, the substrate pocket of rabbit Cma1-like contained the amino acids Ser_189_-Ala_190_-Thr_216_-Val_226_, indicating a pocket markedly different from other hematopoietic serine proteases. Of interest, the replacement from Gly for classical chymases into Thr in position 216 might change the properties of the binding pocket. Since position 216 was predicted to be in the neck of the pocket, Gly216 formed a wide and non-polar hydrophobic entrance, while Thr216 gave the pocket a narrow entrance with hydrophilic properties. Also, because of the different molecular structures in the side chains between Gly and Thr, the substrate pocket of rabbit Cma1-like was likely to be narrower than the classical chymases with approximately equal depth.

In order to study the effects of amino acid differences in position 216 from Gly to Thr, we used the program Phyre 2 to predict a 3D structure of rabbit Cma1-like. The human chymase (human Cma1) was chosen as a reference structure to model the rabbit enzyme and to compare the similarities and differences to the rabbit enzyme. The crystal structure of the chymase (PDB: 4KP0) have been solved [23], and the human chymase (human Cma1) has been shown to prefer aromatic amino acids, such as Phe, Tyr, and Trp, in the P1 position [24,25,26]. When comparing the structures of the two serine proteases, the most obvious difference was the size of the pocket (Figure 6A,B). For human Cma1, Gly216 in the neck of the pocket enabled a wide opened entrance and a large space for accommodating the side chains of the amino acids in the P1 position. As the smallest amino acid, the side chain of Gly was a single hydrogen atom rather than the classical carbon skeleton. The special property of Gly allowed the pocket to recognize and accommodate large functional groups, such as benzene rings of the aromatic amino acids. It seemed a likely explanation for not only the human chymase but also chymases in other species that select aromatic amino acids as the preference in the P1 position. However, Thr took the place of Gly in position 216 in rabbit Cma1-like at the neck of the pocket. This residue change reshaped the pocket as a narrow but still deep passage, resulting in an altered preference for the P1 position. The additional side chain of Thr was inserted into the pocket, occupying the space, forming a narrower pocket that could no longer accommodate aromatic amino acids. However, Leu seemed to fit into the pocket. The side chain of Leu was a carbon structure without a large functional group, allowing the side chain to pass the narrow neck of the pocket. The side chain of Leu contained four carbon atoms forming a “Y” shaped linear structure, which was long enough to reach the bottom of the pocket, stabilized by various interactions, including hydrogen bonding, hydrophobic effects, and Van der Waal’s forces. The powerful interaction between and pocket and the side chain of Leu in the P1 position, in return, provided essential stabilization for the catalytic center to recognize and cleave the peptide bonds in the substrate.

The measurements of the distance between the atoms confirmed this prediction. Ala190 lied in the bottom of the pocket in the human chymase based on the crystal structure (PDB: 4KP0) analysis, and the 3D structure prediction showed a similar situation. The measurement of the distance between the carbon atoms of the side chain of Ala190 and the main chain carbon of Gly216 showed a distance of 8.6 Å in the human chymase, whereas the distance between the same carbon atom of Ala190 and the second carbon atom in the side chain of Thr216 was 7.2 Å in rabbit Cma1-like (Figure 6C,D). Since the Ala190 was predicted and confirmed in the same position of the two chymases, the measurements showed that the side chain of Thr216 in the rabbit Cma1-like did indeed insert into the pocket, occupying the space forming a narrower pocket in the protease, based on the prediction in Phyre 2.

When we looked at the guinea pig, the position 190 was occupied by a Val instead of an Ala as in the rabbit, and in position 216, we had an Ala instead of a Thr. Ala is smaller than the Thr but larger than Gly, which would indicate that the guinea pig might accommodate larger amino acids in the P1 pocket. However, the combination of Val, which is larger than Ala in position 190, might also contribute to restricting the space of the P1 pocket, giving the enzymes very similar primary and extended specificities, although differences are seen in two of the residues forming the P1 pocket. Both the rabbit and the guinea pig enzymes also have a Val in position 226 compared to a smaller Ala for all the enzymes with chymotrypic specificity, which most likely is of importance for the rabbit and guinea pig elastolytic primary specificity (Figure 1).

## 3. Discussion

Here, we presented detailed analyses of the primary and extended specificities of the rabbit β-chymase and the guinea pig α-chymase. Although the Cma1-like sequence is similar to most other mammalian β-chymases, the rabbit Cma1-like represents a protease with a relatively unique primary specificity, being a very strict Leu-ase. Leu-ase activity is relatively unique among known hematopoietic serine proteases. To our knowledge, the only previously reported is the guinea pig chymase from the lab of George H. Caughey [3,17]. Therefore, the rabbit Cma1-like is the second hematopoietic serine protease to be identified as a strict Leu-ase.

The rabbit Cma1-like is located in the chymase locus, which in humans also harbors the genes for neutrophil cathepsin G and the T-cell granzymes B and H. In the rabbit, this locus contains five protease genes: Cma1, Cma1-like, granzyme H, a duodenase-like gene, and granzyme B (Figure 7). The guinea pig Cma1 belongs to the subfamily of α-chymases, whereas rabbit Cma1-like is a β-chymase (Figure 1A). Interestingly, in most rodents, the original α-chymase has changed specificity due to several mutations that restrict the pocket for substrate binding from a chymase to an elastase [22]. For some rodent species, including the golden hamster, Chinese hamster, and rabbit, only one β-chymase gene has been identified. The golden hamster β-chymase has recently been shown to be a classical chymase, and the α-chymase is an elastase [22]. Here, we concluded that the rabbit enzyme was a strict Leu-ase similarly to the guinea pig serine protease Cma1. It is noteworthy that both enzymes in this study had very similar extended specificities despite originating from different subfamilies within the chymase locus—the β-chymases and α-chymases—respectively. The next fundamental questions arise regarding their in vivo roles and targets, especially relating to the classical rodent MC chymases.

The Leu-ase activity of two rodent chymase locus proteases also puts an additional level of complexity to the large diversity in primary cleavage specificities of the hematopoietic serine proteases encoded from the chymase locus. This locus contains an array of specificities, including chymotryptic, represented by the majority of MC α-and β-chymases; tryptic, as represented by human cathepsin G; elastolytic, seen in the mouse, rat, and hamster α- chymases; and Asp-ase, represented by granzyme B’s.

In light of the major change in the primary and extended specificities of the rabbit Cma1-like, compared to the majority of mammalian MC chymases, the questions concerning in vivo substrates of the rabbit enzyme become particularly challenging. Only two chymase genes are found in the rabbit genome, one of which belongs to the α-chymases and one to the β-chymases (Figure 7). As mentioned previously, the mutations in critical positions for an activity most likely have resulted in the functional inactivation of the rabbit Cma1. The β-chymase, rabbit Cma1-like, has previously been thought to have taken the in vivo role of the rabbit α-chymase, which has been described in other rodents, such as rMCP1, mMCP4, and the hamster chymase 1 [11,13]. However, as we showed here, rabbit Cma1-like appears to be a strict Leu-ase, which means the aromatic amino acids, such as Phe and Tyr, are not potential targets for this protease. The seemingly independent origin of very similar Leu-ase specificities in both the rabbit and guinea pig from a β- and α-chymase, respectively, is particularly interesting and indicates a selective advantage, possibly obtained through convergent evolution. One insightful question is if rabbits and guinea pigs can function without a classical chymase that seems to be conserved throughout mammalian evolution from monotremes, marsupials, and most placental mammals? One possibility would be that an enzyme from another locus is taking over the functions of a missing chymase. Therefore, such functions, including toxin inactivation, Ang II generation, selective cleavage of cytokines, and possibly regulation of connective tissue turnover by fibronectin cleavage, as well as other potential chymase-dependent targets, could be carried out by another protease(s) [29,30]. No obvious candidates have yet been identified. However, an earlier unpublished observation in our lab from the opossum might indicate that this situation is a realistic possibility. The opossum chymase locus contains only two genes, one α-chymase and one granzyme B-like gene, which is in marked contrast to rats and mice (vast chymase locus expansion), which all have very similar living environment [5,24]. Why can the opossum manage with only one chymase and one granzyme B, whereas mice and rats have large numbers of additional β-chymases, mMCP-8, and a number of additional granzymes that seem to have appeared at least partly independently through successive gene duplications? Mice, rats, and hamsters also have an additional enzyme specificity in their MC enzyme repertoire, the α-chymase that has become elastases. The apparent independence of the gene duplications in mice and rats that resulted in these additional genes does, in our mind, indicate an evolutionary advantage of having these additional genes. We, therefore, looked for additional serine protease genes using degenerate PCR primers in the skin and intestine of the opossum. To our surprise, we found several enzymes related to the pancreatic chymotrypsins and pancreatic elastases in these tissues. This might indicate that other loci have been involved in increasing the proteolytic complexity in the skin and intestines of the opossum, compensating for the lack of additional enzymes originating from their chymase locus. A similar situation might have occurred in the rabbit and guinea pig, a possibility that needs further investigation.

## 4. Materials and Methods

### 4.1. Production and Purification of Recombinant Rabbit and Guinea Pig Enzymes

The sequence of rabbit Cma1-like was retrieved from the NCBI genome database, and the cDNA sequence was assembled from the genomic sequence. A cDNA copy of the mRNA was designed, based on the gene sequence, and ordered as a designer gene from GenScript (Piscataway, NJ, USA). The designer gene was inserted in the mammalian expression vector pCEP-Pu2, and the sequence was verified. The construct contained a signal sequence, an N-terminal His-6 tag followed by an enterokinase site (Asp-Asp-Asp-Asp-Lys) for purification and subsequent activation by enterokinase cleavage. The vector was transfected into the human embryonic kidney cell line HEK293-EBNA for the expression of the recombinant enzyme. After purification on IMAC Ni^2+^ agarose, the enzyme was activated by the addition of 1 µl enterokinase into 90 µl of column eluate containing the recombinant protein. The sample was mixed, followed by a 5 h incubation at 37 °C to activate the protease. The purity and activation of the enzyme were determined by separation on 4–12% pre-cast SDS-PAGE gels (Invitrogen, Carlsbad, CA, USA). A 2.5 µl of sample buffer, containing sodium dodecyl sulfate (SDS), and 0.5 µl β-mercaptoethanol were added to 10 µl of protein sample followed by 85 °C heating, before separation on the SDS-PAGE gel. The gels were then stained overnight in a colloidal Coomassie staining solution followed by de-staining by several washes [31].

The coding regions for the guinea pig chymase were retrieved from the Swiss Prot/Tr EMBL database (P85201), inserted into the baculovirus vector pAcGP67B, for expression in baculovirus-infected insect cells, and purified, as previously described in detail [18]. Following purification, the fully mature enzyme (25–28 kDa) was ≥95% pure as determined by SDS-PAGE, and the correct N-terminus was confirmed by N-terminal sequencing.

### 4.2. Determination of Cleavage Specificity by Phage-Displayed Nonapeptide Library

A library of T7 phages was designed to contain a random nonameric peptide followed by a His-6-tag at the C-terminus of the capsid protein. The library contains approximately 50 million independent phages. This library was used to determine the cleavage specificity of rabbit Cma1-like, as previously described [9,32,33]. The 250 µl of nickel–nitriloacetic acid (Ni–NTA) beads were used to immobilize the phages based on the interaction with His-6 tags by mixing and gentle agitation at 4 °C for 1 h. The 1.5 mL of washing buffer (1 M NaCl, 0.1% Tween-20 in PBS, pH 7.2) was then added to remove unbound phages. This was repeated 10 times to ensure proper washing. The beads were then washed two times in 1.5 mL PBS and suspended in PBS in a total volume of 500 µl. An appropriate amount of recombinant protease (plus a negative control) was then added, followed by gentle agitation at 37 °C for 2 h. Phages containing peptide sequences, which are cleavable by the protease, were released from Ni–NTA beads, and by centrifugation, the released phages could be separated from the still un-cleaved and thereby bead-bound phages, enabling the recovery of the cleaved phages from the supernatant. The total of 15 µl Ni–NTA beads were added to the supernatant in order to remove phages that still had His-tags but had been released by other means than cleavage ensuring that the His-6 tags had been removed on all the phages in the supernatant. Then, 100 µl of 100 mM imidazole was used to elute the phages from the remaining Ni–NTA beads to determine the number of phages initially bound. In order to calculate the phages detached from the Ni–NTA beads by the protease, a series of dilutions of the supernatant were plated onto LA-Amp plates together with 3 mL of 0.6% top agarose, 300 µl of *Escherichia coli* (BLT 5615), and 100 µl of 100 mM isopropyl β-D-1-thiogalactopyranoside (IPTG). *Escherichia coli* (BLT 5615) (10 mL) was prepared to amplify the remaining phages for the next round of selection. To this culture, 100 µl of a 100 mM solution of IPTG was added to induce the expression of the T7 phage capsid protein. The bacteria were lysed after approximately 2 h of gentle agitation at 37 °C, and the lysate was centrifuged to remove cell debris before storage at 4 °C for the next round of selection. After 5–7 rounds of selection, 120 plaques were picked from LA-amp plates after plating in top agarose, and the plugs were transferred into tubes containing phage extraction buffer (100 mM NaCl, 6 mM MgSO_4_, 20 mM Tris-HCl, pH 8.0). Vigorous shaking for 30 min at 4 °C was critical for extracting the phages from the agarose. The polymerase chain reaction was then used to amplify the phage DNA encoding the random nonamer region. The quality and quantity of the amplified DNA were determined by gel electrophoresis, and the 96 samples with the best DNA quality were sent in a microtiter plate for sequencing to GATC Biotech (Ebersberg, Germany).

### 4.3. Generation of Recombinant Substrates for the Analysis of the Cleavage Specificity

To verify the result from the phage display, we used a new type of recombinant substrate. Two copies of the *E*. *coli* thioredoxin (trx) gene were inserted in tandem into the pET21 vector for bacterial expression (Figure 4A). In the C-terminal end, a His-6 tag was inserted for purification on Ni^2+^ IMAC columns. In the linker region, between the two trx molecules, the different substrate sequences were inserted by ligating double-stranded oligonucleotides into two unique restriction sites, one BamHI and one SalI site (Figure 4A). The sequences of the individual clones were verified after cloning by the sequencing of both DNA strains. The plasmids were then transformed into the *E. coli* Rosetta gami strain for protein expression (Novagen, Merck, Darmstadt, Germany). A 10 mL overnight culture of the bacteria, harboring the plasmid, was diluted 10 times in LB + Amp and grown at 37 °C for 1–2 h until the OD (600 nm) reached 0.5. IPTG was then added to a final concentration of 1 mM. The culture was then grown at 37 °C for an additional 3 h under vigorous shaking, after which the bacteria were pelleted by centrifugation at 3500 rpm (or RCF 2851) for 12 min. The pellet was washed once with 25 mL PBS + 0.05% Tween 20. The pellet was then dissolved in 2 mL PBS and sonicated 6 × 30 s to open the cells (Soniprep 50, Amplitude micron 30, max power and max tune). The lysate was centrifuged at 13,000 rpm for 10 min, and the supernatant was transferred to a new tube. Ni-NTA slurry (0.5 mL) (50% slurry concentration) (Qiagen, Hilden, Germany) was added, and the sample was slowly rotated for 45 min at RT. The sample was then transferred to a 2 mL column, and the supernatant was allowed to slowly pass through the filter, leaving the Ni-NTA beads with the bound protein in the column. The column was then washed four times with 1 mL of washing buffer (PBS + 0.05% Tween + 10 mM Imidazole + 1 M NaCl). Elution of the protein was performed by adding 150 µl elution buffer, followed by five 300 µl fractions of elution buffer (PBS + 0.05% Tween 20 + 100 mM imidazole). Each fraction was collected individually. Samples (10 µL) from each of the eluted fractions were then mixed with 1 volume of 2x sample buffer and 1 µL β-mercaptoethanol and then heated for 3 min at 85 °C. The samples were analyzed on an SDS bis tris 4–12% PAGE gel, and the second and third fractions that contained the most protein were pooled. The protein concentration of the combined fractions was determined by the Bio-Rad DC Protein assay (Bio-Rad Laboratories, Hercules, CA, USA). Approximately, 25 µg of recombinant protein was added to each 50 µL cleavage reaction (in PBS). A sample (10 µL) from this tube was removed before adding the enzyme for the 0-min time point. The active enzyme was then added, and the reaction was kept at room temperature during the entire experiment. Approximately, 3 µg of the guinea pig enzyme and 0.8 µg of the rabbit enzyme were used for a 50 µl reaction. Samples (10 µl and approximately 5 µg of protein) were removed at the indicated time points (15, 45, and 150 min), and the reaction was stopped by the addition of one volume of 2× sample buffer. β-mercaptoethanol (1 µL) was then added to each sample followed by heating for 3 min at 85 °C, and 15 µl from each sample was then analyzed on 4–12% pre-cast SDS-PAGE gels (Invitrogen, Carlsbad, CA, USA). The gels were stained overnight in the colloidal Coomassie staining solution and de-stained, according to the previously described procedure [31].

## Figures and Tables

**Figure 1 ijms-20-06340-f001:**
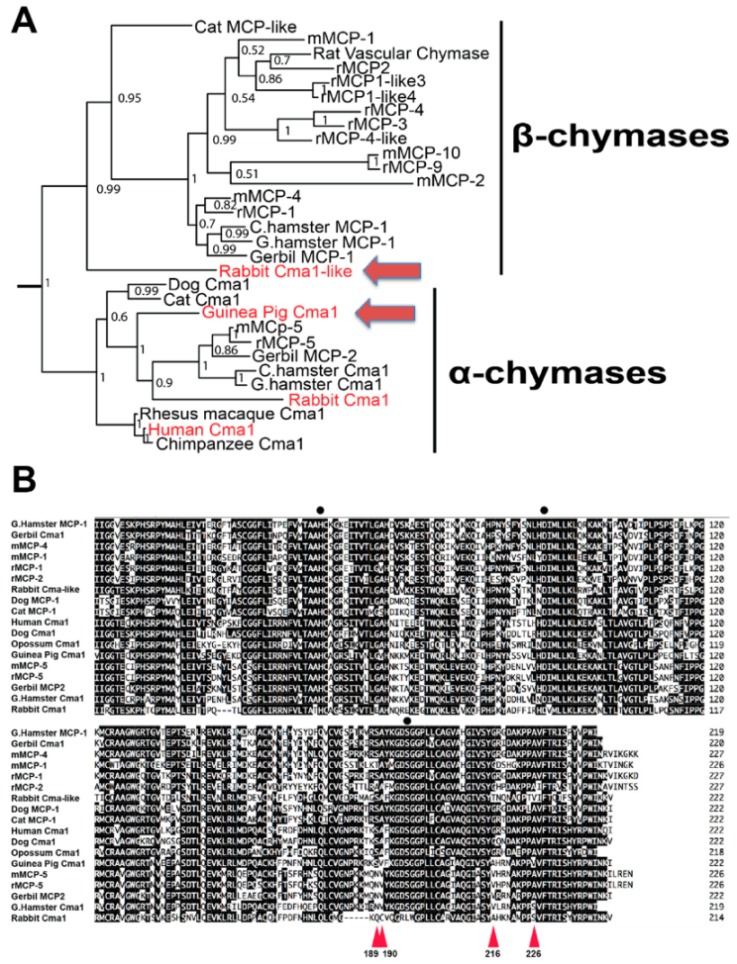
A phylogenetic tree of selected mammalian chymase locus genes, and alignment of a panel of mammalian α and β-chymases. (**A**) shows a phylogenetic tree based on the relationships among chymase locus genes of a number of mammals using both MrBase analysis program and maximum-likelihood algorithm [5]. The enzymes of major interest for this study are marked in red, and the two enzymes analyzed in detail are marked by large red arrows. (**B**) shows an alignment of a panel of mammalian α-chymases and β-chymases. The positions of the three residues of the catalytic triad His-Asp-Ser are marked by black dots. The four amino acids, which were predicted as lining the substrate pocket, were numbered 189, 190, 216, and 226, as indicated by red triangles [5,10,18,19]. The DNA Star Megalign program and Clustal W algorithm were used to create the alignment [20].

**Figure 2 ijms-20-06340-f002:**
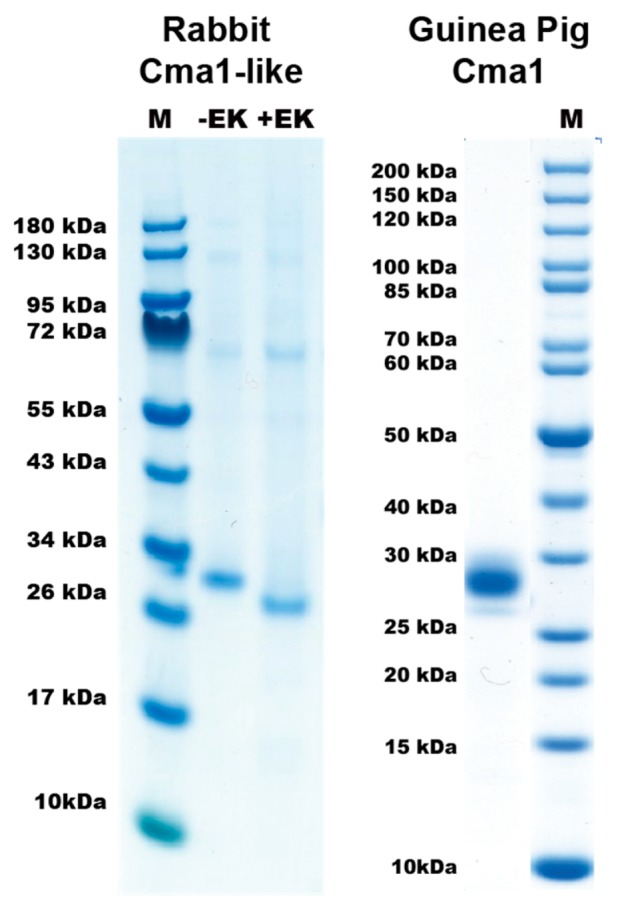
SDS-PAGE of the rabbit and guinea pig enzymes used in this study. The rabbit Cma1-like was produced as an inactive enzyme containing an N-terminal His-6 tag and an enterokinase site (EK). The enzyme was produced in the human cell line HEK293-EBNA with the episomal vector pCEP-Pu2. Enterokinase was used to cleave off the N-terminal to activate the chymase. The inactive enzyme with the N-terminal purification tag and the active enzyme were analyzed by separation on SDS-PAGE and visualized with Coomassie Brilliant Blue staining. PAGE Ruler was used as a molecular weight marker. M: marker; +EK: with enterokinase; −EK: no enterokinase. The guinea pig chymase was produced in insect cells with a baculovirus-based expression system [18].

**Figure 3 ijms-20-06340-f003:**
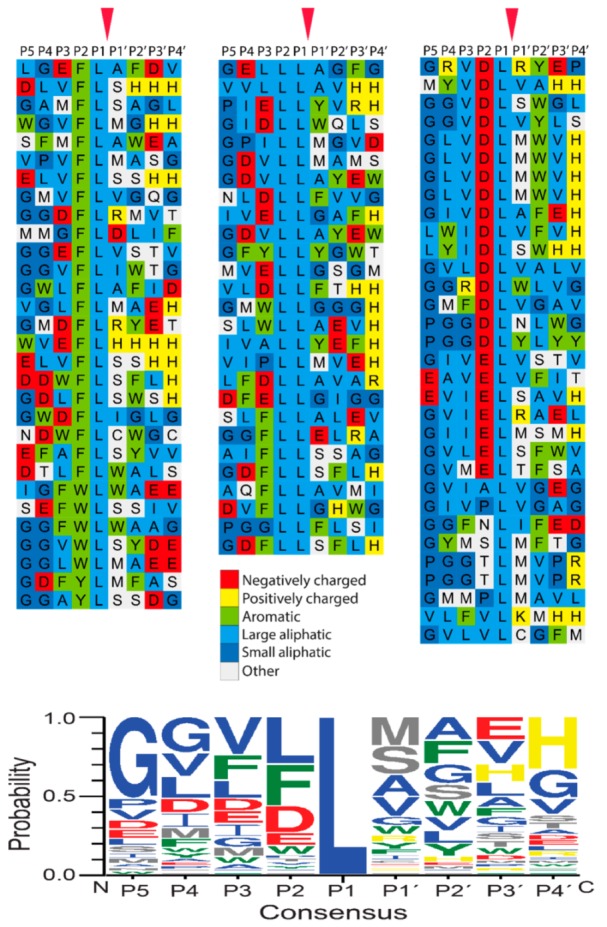
Phage display analysis of rabbit Cma1-like chymase after six rounds of selection. After the last round of selection, the released phages were collected for sequencing. The general amino acid sequences are P-G-G-X-X-X-X-X-X-X-X-X-H-H-H-H-H-H, where X denotes to randomized amino acids. The amino acid sequences were aligned into a P5-P4′ consensus. The amino acids were color-coded based on the side chain properties, as shown in the middle of the figure. At the bottom of the figure, we showed an Ice-Logo type presentation using the Web-logo program of the distribution of amino acids in positions P5 to P4′ based on the phage display alignment by rabbit Cma1-like chymase after six rounds of selection [21]. Amino acid groups are colored as in the phage display sequences depicted in the upper part of the figure, except for the amino acids, Ser, Thr, and Met, which are depicted in white at the top of the figure. They are shown in grey in the Web-Logo part of the figure.

**Figure 4 ijms-20-06340-f004:**
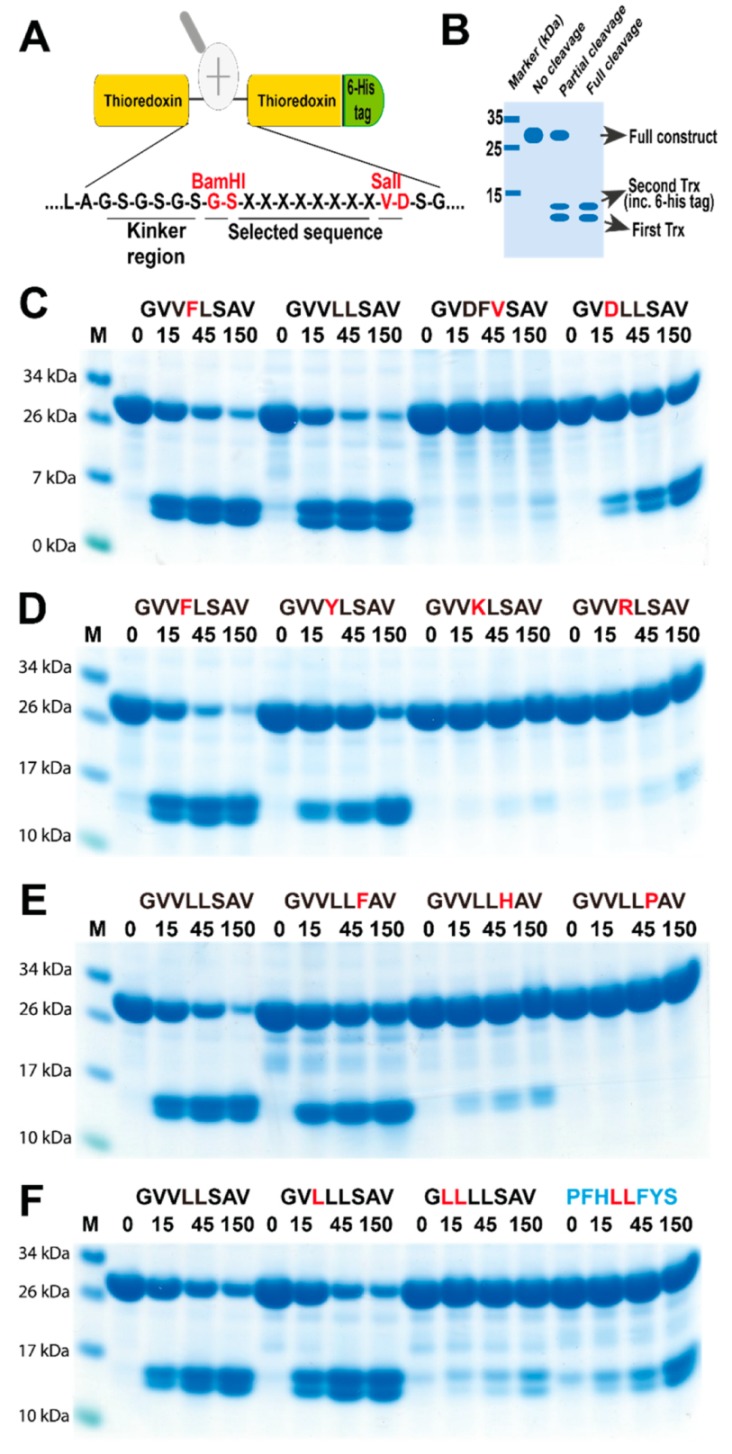
Analysis of the cleavage specificity of rabbit Cma1-like chymase with recombinant protein substrates. (**A**) shows the overall structure of the recombinant protein substrates used for the analysis of the efficiency in cleavage by the rabbit Cma1-like. In these substrates, two thioredoxin molecules were positioned in tandem, and the proteins had a His-6 tag positioned in their C termini. The different cleavable sequences were inserted in the linker region between the two thioredoxin molecules by the use of two unique restriction sites, one Bam HI and one SalI site, which are indicated in the bottom of panel A. Within the linker region, there was a flexible kinker region consisting of repeated Gly and Ser residues. (**B**) an example cleavage is shown to highlight possible cleavage patterns. (**C**–**F**) show the cleavage of a number of substrates by rabbit Cma1-like. The sequences of the different substrates are indicated above the pictures of the gels. The time of cleavage in minutes is also indicated above the corresponding lanes of the different gels. The un-cleaved substrates had a molecular weight of ~25 kDa, and the cleaved substrates appeared as two closely located bands with a size of ~13 kDa.

**Figure 5 ijms-20-06340-f005:**
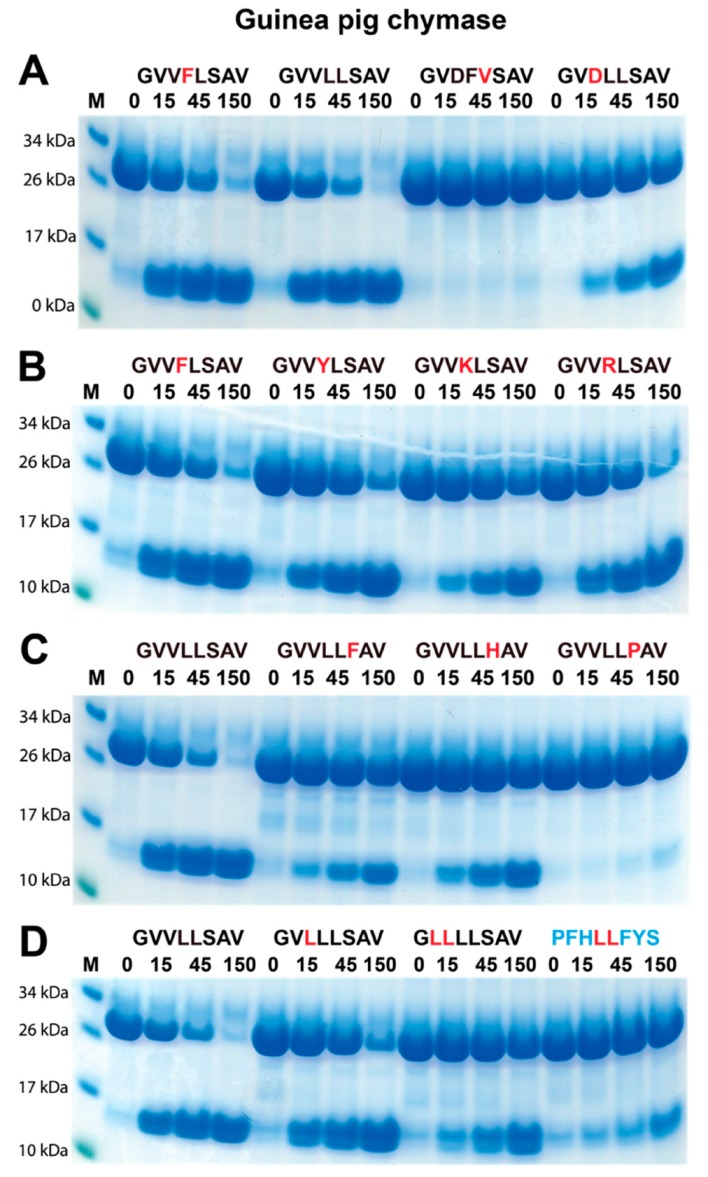
Analysis of the cleavage specificity of guinea pig Cma1 chymase with recombinant protein substrates. (**A**–**D**_**A**; **B**; **C**; **D**; **C**; **D**; **E**; **F**;_) show the cleavage of a number of substrates by guinea pig Cma1. The sequences of the different substrates are indicated above the pictures of the gels. The time of cleavage in minutes is also indicated above the corresponding lanes of the different gels. The un-cleaved substrates had a molecular weight of ~25 kDa, and the cleaved substrates appeared as two closely located bands with a size of ~13 kDa.

**Figure 6 ijms-20-06340-f006:**
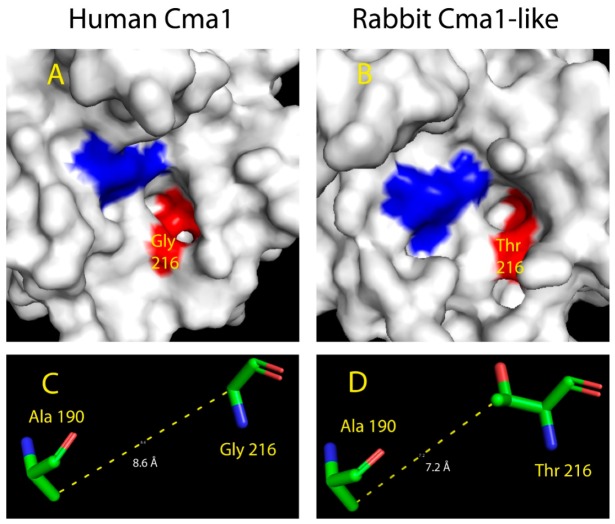
Modeling of the substrate-binding pockets and measurement of distances in human Cma1 and rabbit Cma1-like chymases. Human Cma1 (PDB: 4KP0) was chosen as the example of a chymase structure. The website of Phyre 2 was used to predict the 3D structure of rabbit Cma1-like [27]. The sequence of rabbit Cma1-like was retrieved from the NCBI database. The software PyMol was used for the analysis of the 3D structures [28]. (**A**,**B**) show the surface structures of the chymases. The positions of the three residues of catalytic triad His-Asp-Ser are colored in blue. The positions of the four amino acids of the substrate pocket, 189-190-216-226, are colored in red. Locations of Gly216 and Thr216 were identified. (**C**,**D**) show the distance between Ala190-Gly216 and Ala190-Thr216, respectively.

**Figure 7 ijms-20-06340-f007:**
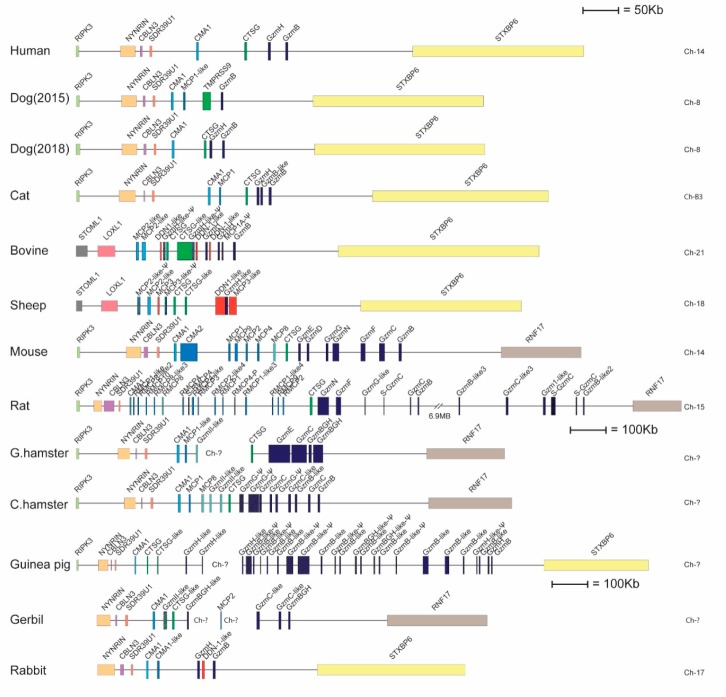
The chymase locus. The chymase locus encodes a number of hematopoietic serine proteases, including the α-chymases, β-chymases, cathepsin G, and several granzymes [5]. In ruminants and rabbits, there is also a new subfamily of proteases, the duodenases. Genes are color-coded, the α-chymase-related genes are marked in light blue, the β-chymases in slightly darker blue, cathepsin G in green, the M8 family in a darker green, the granzymes in dark blue, and duodenases in red.

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
