# Peer review of "Extended Cleavage Specificities of Rabbit and Guinea Pig Mast Cell Chymases: Two Highly Specific Leu-Ases"

_ijms, 2019, doi:10.3390/ijms20246340_

Round 1

Reviewer 1 Report

Zhongwei et al. analyze the substrate specificity of the rabbit and guinea pig MC chymases; though the work is scientifically sound I need to state that a very central part is missing: the identification of a (putative) native substrate either from rabbit or guinea pig. For the purpose of the in vivo substrate identification of either proteases a N-terminal, MS-based substrate identification method like HUNTER, TAILS or CoFRADIC would be the ideal solution.

Or in case no MS instrumentation would be easily available for the authors (though this would be a much better way to go), I would suggest to grep a bit more on the native substrates for the chymases: potentially by finding homologous sequences to the identified “better” cleaved recombinant protease substrates in the guinea pig or rabbit proteome. Or at least postulate some candidate proteins from the rabbit or g.pig proteome which would need to be cleaved in the corresponding assays with the chymases Leu-proteases to identify putative targets.

In addition to that, the figures in the paper need a bit of “compressing” the information into less panels or visual elements to improve the legibility of the paper – but, maybe this partially also arose from the submission system and the automatic conversion of the manuscript into the PDF. Besides this, there is sometimes redundant information present in different figures/panels: for example, the cleavage size specificity could be displayed in a much better manner using iceLogos (https://iomics.ugent.be/icelogoserver/create) from the cleavage windows which would combine figures 3 & 4 (iceLogos normalize the amino acids present at the position to the corresponding natural abundance). I would strongly suggest the authors to display the specificity in this manner to improve readability and value of information.

The paper is nice but the figure representation needs to be improved a lot and if the authors could provide a (putative) physiological substrate, I feel the story would be quite nice to read.

Specific comments:

Figure 1A: smaller fonts for the dendrogram + smaller red arrows Figure 1B: this needs a major re-design: please avoid the dark black background and apply a grey’ish background, additionally emphasize the substrate pocket amino acids by a rectangular box each. Figure 2: make this figure much smaller, the information deduced from this figure is not that high and thus should not occupy so much space in the manuscript. Why both chymases were expressed in different expression systems? And why was not the complete protease sequence determined by MS to be absolutely sure, that the proteases were completely intact (e.g. to rule out single site mutations rather at the C-terminus?)? Figure 3&4: combine both into an iceLogo – if you want to keep to the current state of these figures, then these need a little of a overhaul: Figure 3: separate P1 – P1’ and indicate the scissile bond more clearly (maybe dashed line?), take more pale’ish colors and try to avoid green & red in one figure please. I would also split these figure in a 4 column layout to save again space and to display the elements in a more compacted form. Figure 4: Far too much whitespace, this could be also presented in a 3- or 4-column layout. L220-230: Important points, but these sections should be re-phrased to be more compacted. Figure 5: Why was this Bi-Trx system chosen to express the recombinant substrates? Any rationale should be stated more obvious in the main text or the figure legend. 5B: The arrows are far too faint to be recognized, maybe just use lines to align with the text. 5C…: Protein amount or volume load for these gels should be mentioned in the legend. I would also like to have an indicator besides the double band to re-align the First/Second Trx with the “legend” figure 5B. Is a Western Blot of the His-tagged First Trx samples available? At least for one exemplary experiment I would display the clear His-tagged construct localization/size on a western blot. Figure 5/6 in general: could these figures be merged? And if not – what about a quantification of the remaining, full length substrate over the time course for different substrates? Would be an additional means to quantify the “quality” of the different recombinant substrates. L269-L292: Rephrase the section, too lengthy in the current shape! L290: between and pocket Figure 7: Would depicting the distances from panels C/D be possible directly within A/B? And could a putative L or F/Y residue be placed inside the surface/distance display? L360…: The jumping on the opossum train at this section is quite irritating and not appropriate in the current manuscript entitled rabbit/g.pig chymases. Figure 8: Remove the background shade. L395: What would be the corresponding amount of protein for these 10 µL? L436: Why the need for manual alignment? L455: 3500 rpm = how much g? The rotor is not stated anyways, so give the exact g centrifugal force please. L456: sonication settings and instrument? L466 / 474: ß missing or too much space within “ß-mercaptoethanol”

Throughout the methods section, the denaturation seems to be performed sometimes at 80 °C and sometimes at 85 °C – was this really the case, different temperatures were used?

Author Response

Reviewer 1.

The question concerning potential in vivo substrates for the rabbit and guinea-pig Leu-ases.

Response; We fully agree that it would be great to know the in vivo substrates for these two enzymes. This especially as they represent a relatively unique primary and extended specificity to mast cells, contrasting to all other species analyzed from monotremes to humans that have classical mast cell chymases. However, this is not an easy task to be completed with a time frame of 6 months, or even several years and certainly not for the 7 days set for the revision.  

                      After more than 30 years of intensive studies and probably more than 100 articles on the human chymase we still only have a few reasonably well-established targets for this enzyme. Angiotensin I is most likely an important substrate as well as several TH2 cytokines, several snake and scorpion toxins and fibronectin. However, concerning most other potential targets described in the literature it is very uncertain if they are true in vivo targets.

                      The phage display results are very clear that the rabbit enzyme is a strict Leu-ase. However, the screening of the rabbit genome with this consensus sequence would most likely result in several thousand hits similar to the analysis we have performed with the consensus sequence obtained for the human chymase. On top of that, as we could see in our recent JI article on the cleavage of human cytokines and chemokines by the human mast cell chymase, even if the consensus site is present in the primary sequence, it is often observed that the site is not a target when the protein is properly folded.  This due to that the site is hidden in the structure and not accessible for the enzyme to cleave. We can speculate about potential targets but have no good suggestions at the moment. Whole tissue analysis using MS or 2D gels are also very difficult to evaluate as many potential substrates can appear in tissue extracts of which the absolute majority are not biologically relevant substrates as they either are cytoplasmic or nuclear proteins that never will come in contact with the enzyme in vivo and many are structural or plasma proteins present at very high concentrations and in the presence of protease inhibitors where the cleavage will have very little biological effect. We certainly agree that the question of the major in vivo substrates for these two enzymes are a top priority but it is a very difficult issue that needs several years of work to sort out what is in vivo relevant and not. The problem is also that very few pure proteins and reagents are available for rabbit or guinea pig that makes an analysis in these two species an order of magnitude more time consuming and difficult compared to the analysis of mouse and human proteases. The studies we have performed on cytokine and chemokine cleavage by human and mouse mast cell enzymes would not be possible to perform on Rabbit and guinea pig due to the lack of the majority of these potential target proteins in these species. In conclusion we totally agree that this is a very interesting issue that needs to be addressed but it is out of the scope of this manuscript as it would need several years of intensive work, in combination with a high risk of not succeeding in identifying the key biologically most important targets, similar to what has been seen for both the human and the mouse mast cell chymases.

Concerning figures;

We have now edited figure 1 to reduce font and arrows. The sizes of the figures are determined by the journal production office that we have little impact on but we agree that they can easily be reduced in size. Concerning the black to grey suggestion we think that should reduce readability of the figure as the text is now in white which would be more difficult to see in a grey background. To introduce a box for the residues of interest should also shield the text and make it less readable.

We have now combined figures 3 and 4 and deleted figure 4 and replaced it by a variant of the iceLogo using the Weblogo program that we feel is much better and more user-friendly.

We have also added red arrows showing the actual cleavage site in the sequences.

The arrows in former figure 5 (present figure 4) has now been increased in size as suggested by the reviewer. We also want to point out that this part of the figure (Panel B) is just a schematic illustration and not a western blot.

Concerning color. We have published the extended cleavage specificity of a large number of other hematopoietic serine proteases and there used this color coding. In order to be consistent and deliver comparative figures we prefer to keep this color coding.

Figures 5 and 6 (new 4 and 5) are difficult to merge into one page as it would be confusing for the reader.

Figure 6 former 7 ; to introduce both distance and residues in the same figure would not save space as the figure still would take up the same space in the final document and it would be more crowded and difficult to follow. The exact position of the leu would also not be easy to determine and would shield the position of the other key amino acids.

Concerning figure 8 present 7 there is no dark background only white. Possibly the web program used by the reviewer resulted in such an effect. When I opened the submitted manuscript the background on the submitted version was white and will be white in the final article.

Concerning the enzymes;

The guinea pig enzyme was produced by Jukka Kervinen and Larry de Garavilla then at Johnson and Johnson. We had it in our freezer together with 7 other mast cell proteases produced by them, not including a rabbit enzyme. As we do not have the insect expression system going and it needs a lot of time and effort to set it up we produced the rabbit enzyme in the mammalian system. Both show good activity when tested with the optimal substrates but have no activity with any of the chromogenic substrates tested.

The sequences of the clones were tested before transfection and not found to have any point mutations. Single amino acid mutations in a small fraction of the protein if mutated after transfection in some cells would not be detected by MS or any other protein method and would also not affect cleavage results as its such a small fraction of the protein if at all present.

Concerning the use of the 2xTrx system;

We have now added small section to the results section where we describe the reason why we use this system (marked in red). It is an inhouse system that has proved to be one of the best systems (probably the best) available to study the cleavage preference for different endopeptidases. It is very visual, transparent and easy to use and do not need fancy equipment and no need for complicated and expensive organic chemistry synthesis. As we also state in the manuscript we have now produced more than 270 such proteins in a standard molecular biology lab without any expensive machinery and used it to obtain detailed quantitative information about most of the mammalian hematopoietic serine proteases presently characterized. References for many of these previous studies are also listed in the manuscript. This system has worked much better than we in our wildest fantasy could have imagined when I come up with the idea, a few years ago. We have also observed that several proteases that do not cleave conventional chromogenic substrates due to that they lack amino acids C-terminal of the cleavage site shows good cleavage with the 2xTrx substrates.  

I think it is great that we have developed as system that is so simple and easy to use and so visual that it can be used in a student lab and still get the best results that can be obtained with any technology presently available. The only problem we have observed is when the target sequence consists of three or more basic amino acids in a row. Then there is substantial cleavage within the bacterial cells where the substrate is produced giving cleavage product also at the 0 minute starting point. Not so nice to look at but still possible to use.

Concerning the opossum results;

We disagree on this point as the opossum is the only species where we can have some potential in vivo indication for the last statement in the abstract-that another locus is acting to rescue the lack of chymotryptic enzymes in the rabbit and guinea pig. It is in our mind a very important piece of information to put the rabbit and guinea pig results in a bigger perspective. I can here mention that we are presently working on the cow and sheep duodenases, which is a second possible example where a similar mechanism may be involved. There are a few duplicated copies of a cathepsin G or granzyme gene within the chymase locus in cows and sheep that now are acting as digestive enzymes and have changed tissue specificity from hematopoietic cells to now instead being expressed in the duodenum. We therefore feel that this information obtained from the opossum studies is a very valid information and important for the understanding of the evolutionary processes that act on these proteases and on the genomes in general.

Concerning the amount of protein in 10 ul;

It is 5 ug which has been added to the manuscript (marked in red).

RCF, sonication settings and denaturing temperature have been added and/or corrected (marked in red).

All the changes in the text have been marked in red.

We have now addressed the absolute majority of issues put forward by the reviewers. Although not agreeing with all the suggested changes to the manuscript we hope I have given good explanations why we not fully agree on the level of complexity to get the in vivo substrates and the importance of evolutionary focus, which we feel is the essence of the paper, and also hope this manuscript now is suitable for publication in this special issue of IJMS.

Sincerely

Lars Hellman

Reviewer 2 Report

In the paper entitled “Extended cleavage specificities of rabbit and guinea pig mast cell chymases: Two highly specific Leu-ases”, Yuan Zhongwei et al. report on the cleavage specificity of the rabbit Cma1-like chymase. They performed a phage display analysis to identify the preferred cleavage site of the enzyme and verified their findings using recombinant proteins and an in vitro assay. The main finding of the paper is that the rabbit Cma1-like protein prefers Leu at position P1, which makes this protein the second example of a hematopoietic serine protease with strict Leu-ase activity. In order to explain this specificity, the authors used homology modelling of the rabbit Cma1-like protein using the human Cma1 as a template. They found that one critical aminoacid at position 216 may confer cleavage specificity.

In my opinion, the biochemical demonstration of the cleavage specificity of the rabbit Cma1-like chymase is solid and for this reason the manuscript deserves further consideration. I would nevertheless recommend the authors to improve two aspects of their manuscript as described below.

Regarding the presentation of the work. I think the scientific question is not properly presented in the abstract, the introduction, and the discussion. The main message of the paper is the cleavage specificity of the rabbit Cma1-like protein. Therefore, all the description of the genomic information should be kept to the minimum. As a striking example, the discussion is almost entirely dedicated to speculating around the evolution of the different chymase activities, which in itself is really interesting. However, I believe the paper would rather deserve a discussion about the results presented in this manuscript. For instance, no mention of the homology modelling or the importance of the residue 216 is present in the discussion. I have thus the impression that the introduction and discussion are somehow disconnected from the actual original data presented in the manuscript.

Regarding the homology modelling. I am wondering if the presentation of the guinea pig alpha chymase data are justified in this manuscript. Although the observations are interesting, I have the feeling that they do not serve the main argument. For example, the homology modelling has been performed with the rabbit Cma1-like chymase only and not the guinea pig protein. It is my understanding that the guinea pig protein has a Ala in position 216 but this protein also has a strict Leu-ase activity. Therefore, the argument presented here, that the Thr216 of the rabbit Cma1-like chymase occupies a volume at the neck of the pocket does not seem to stand for the guinea pig protein. Would it make sense to discuss more about the importance of the combination of the two residues 190 and 216 rather than 216 alone? Is it feasible to do the modelling with the guinea pig protein as well? Please note that I am not asking the authors to remove the data about the guinea pig protein but rather to think about this possibility. In the case they want to keep the data in the manuscript, I would expect to see the homology modelling for this protein and a discussion about the data.

Other points:

L60: “uncertainty about the quality of the dog genome sequence” Please provide a reference L69-73: please provide a reference for the data about the wolf DNA. Figure 1A: the size of the font is slightly too big in my opinion making the tree difficult to read. However, I am not asking the authors to change it if they think it is better as it is. Figure 1: please provide reference for the ClustalW algorithm. Figure 2: please add a dashed line for the guinea pig Cma1 purification gel to make it clear the gel has been cut.

Materials and methods:

- please provide the NCBI genome database ID for the rabbit Cma1-like and the guinea pig chymase genes.

- please indicate the amount of enzyme used in the enzymatic assays (4.4).

- please add a section to explain how the modelling has been performed and add references for Phyre2 and PyMol.

Author Response

Reviewer 2

As described for reviewer 1 -We have now edited figure 1 to reduce font and arrows. 

Concerning the focus on gene evolution in major parts of the manuscript.;

In my mind this study only make sense in the light of evolution, ( I also think almost all studies of biology and in many cases also medicine only make sense in the light of evolution) and as we all know the genes are the major players in this evolutionary process. All other studied mammals, from the egg-laying monotremes to humans, have classical mast cell chymotryptic enzymes except these two species. How do they manage to solve the tasks performed by the mast cell chymases in other species? This puts the focus on the role of these mast cell enzymes in humans and mice. Is angiotensin I cleavage one of the main tasks of the human chymase and in that case do rabbit and guinea pig have other angiotensin converters?  Rats have most likely solved the question of a poor angiotensin converting mast cell chymase (rMCP-1) by a gene duplication within the chymase locus giving rise to the rat specific vascular chymase. This is in my mind actually one of the best pieces of evidence for the importance of the human mast cell chymase as angiotensin converter.  As mentioned above we have studied new enzymes of the chymase locus in ruminants and then looked into another immune locus that has experienced major changes, the lysozyme c locus, where humans have one gene, mice have two and cows have 10, and where 4 of them are taking part in food digestion. This shows the extreme flexibility and plasticity of mammalian genomes to solve the problems of changing environment, changing food sources and other important changes in their life style. Studies of the evolution of different biological processes and enzymes is key to our understanding of fundamental processes in biology in general and also in human physiology.  

The focus of this article is really the evolutionary processes that act on mast cell enzymes and their function why gene and gene evolution is central in the manuscript.

Concerning modelling of the guinea pig and the references concerning the rabbit structural analysis;

We have added references to the ClustalW, Phyre2 and  PyMol programs and algorithms. We describe in the text also the way the modelling has been performed. We use the human structure as backbone to model the rabbit amino acid sequence onto this backbone, which results in a relatively good prediction as the enzymes are closely related in overall structure.

As correctly pointed out by the reviewer the sequence of the specificity determining residues of rabbit and guinea pig differ quite markedly. We have now added a section to the results section on the potential effects of these differences and why these two enzymes still have very similar extended specificities. We have no perfect explanation but try to give a possible reason for this similarity in extended specificity despite their differences in sequence.

Concerning the lack of discussion around the structural differences between the chymases and Leu-ases;

The homology modelling gives additional structural information on the change in primary specificity.  I do not know what more interesting we can discuss around this effect except in the light of evolution and why we have this seemingly converted evolution of two enzymes, one alpha and one beta chymase evolving into having a very similar extended specificity. It is not possibly to tell why he enzyme has mutated and the sequence is what it is.  I think the interesting thing is here not the structure itself but the functional significance of this change in specificity, which we discuss in detail in the discussion section. We have now also added a short section of another interesting addition to the immune repertoire of several rodents- the change in primary specificity of the alpha -chymase in rats, mice and hamster which has become elastases. What are the new functions performed by these enzymes and why are they only found in a subfamily of rodents and not in other mammals? That is an additional interesting difference between these rodents and the opossum which has a very similar life style but which lacks the additional chymases found in rats and mice and also lacks this elastase. So there are still several questions concerning the functional significances of these major differences in protease repertoires between relatively closely related species, questions that are important to solve but not easy to tackle as several of these enzymes have a relatively broad specificity and thereby most likely several overlapping functions at different stages in an inflammatory response and most likely also taking part in normal tissue homeostasis.

Concerning the dog genome;  There is no reference just that we have observed major differences when looking at this locus at different time points as the genome sequence constantly are being updated. In an early version of the genome sequence the beta chymase was present and in a later update it was gone and instead a granzyme H gene had appeared which shows that care needs to be taken when looking at the supposedly complete or almost complete genomes. We have also never published the sequence of the beta chymase obtained from an analysis of wolf DNA we have only mentioned the result in a previous publication, it is identical to the beta chymase gene sequence previously found in the dog genome sequence that now has been removed, but that we know is there. As the exact same sequence was present in the genome in the database we found no need to publish it. However, this earlier publication where the wolf sequence is mentioned has now been included as reference.

The concentration of the enzymes used have been added and the accession numbers for the  guinea pig sequence has also been added to the materials and methods section. The rabbit sequence was instead assembled from the genomic sequence why no direct accession number can be added.

The gel picture of the guinea pig chymase is coming from a gel containing two additional hamster proteases not being the focus for this analysis and that has been published recently. To ensure the reviewer this is a correct representation I here include this gel as a png file. See additional PDF file as PNG files apparently can not be included in this system.

  Guinea pig (GP) Hamster chymase 1 (HC1), hamster chymase 2 (HC2).

All the changes in the text have been marked in red.

We have now addressed the absolute majority of issues put forward by the reviewers. Although not agreeing with all the suggested changes to the manuscript we hope I have given good explanations why we not fully agree on the level of complexity to get the in vivo substrates and the importance of evolutionary focus, which we feel is the essence of the paper, and also hope this manuscript now is suitable for publication in this special issue of IJMS.

Sincerely

Lars Hellman

Round 2

Reviewer 1 Report

With the explanatory statements from the lead author I feel now that the paper is quite suitable for publication. Maybe I just missed to encounter the 2Trx-system until now..

Just a small point: regarding Figure 3 - if possible - do not stretch the figure to produce distorted text on the axes. but the Logo is definitely much better to present the central cleavage motif of the rabbit Leu-chymase.